# “We Are Not Different than Others”: A Qualitative Study of the Lived Experience of Hispanic Adolescents and Young Adults Living with Perinatally Acquired HIV

**DOI:** 10.3390/ijerph20042996

**Published:** 2023-02-08

**Authors:** Cynthia Fair, Leandra Fraser, Mackenzie Zendt, Maria Santana-Garces, James Homans, Alice Stek, Eva Operskalski

**Affiliations:** 1Department of Public Health Studies, Elon University, Elon, NC 27244, USA; 2Department of Infectious Disease, University of Southern California Medical Center, Los Angeles, CA 90033, USA

**Keywords:** Hispanic, adolescents and young adults, perinatally acquired HIV, lived experience

## Abstract

Though Hispanic youth with perinatally acquired HIV (PHIV) comprise 14% of those living with PHIV, little research has documented their lived experiences. Eighteen Hispanic adolescents and young adults (AYA) with PHIV were recruited from two pediatric infectious disease clinics in California (mean age = 20.8 years, 12 females and 6 males). Interview transcripts were analyzed for emergent themes regarding relationships, childbearing intentions, and future career aspirations. Participants acknowledged HIV as cause for rejection and fear of transmission from partners. Most desired children in the future. Those with children (n = 7) expressed a strong desire to continue their education for the benefit of their children. Many did not view HIV as a barrier to their career aspirations. HIV influenced their daily lives. However, the challenges of poverty, loss, and trauma also significantly shaped their well-being. Health care providers offered emotional and instrumental support which helped AYA make progress towards their goals.

## 1. Introduction

Approximately 9400 individuals in the United States live with perinatally acquired HIV (PHIV) [1]. Due to increased access to antiretroviral treatment in the last 20 years, youth with PHIV are now living into young adulthood [2]. Today, adolescents and young adults (AYA) with PHIV are facing many of the same issues associated with relationships as their uninfected peers, but the developmentally normative tasks of exploring sexuality are made more complex given that HIV is stigmatized and sexually transmitted [3,4]. Research has investigated how PHIV affects adolescents’ romantic relationships, reproductive health behaviors, and future childbearing desires and intentions [4,5,6,7,8,9]. However, few studies have examined AYA living with PHIV and its effects on relationships and reproductive decision making by race or ethnicity.

Specifically, little is known about Hispanic adolescents who have been living with HIV since birth, who make up approximately 14% of all diagnosed pediatric HIV cases in the U.S. [10]. Research has indicated that aspects of Hispanic culture may influence disclosure patterns of Hispanic youth with behaviorally acquired HIV. D’Angelo et al. found that Hispanic adolescents were less likely to inform their mothers of their HIV status than African American teens [11]. The authors hypothesize this could be due to discomfort admitting risky behaviors because of cultural norms. However, little is known about their counterparts living with PHIV.

Hispanic youth, in general, are less likely than their peers to have consistent access to medical care and culturally sensitive sex education, both of which are vital to healthy sexual and relationship development [12]. Indeed, the Youth Risk Behavior Survey found that Hispanic high school students were more likely to use no pregnancy prevention methods compared to non-Hispanic white students [13]. This project describes the understudied lived experience of Hispanic adolescents with special attention to their relationship experiences, childbearing intentions, and career aspirations.

## 2. Materials and Methods

### 2.1. Participants

This cross-sectional study included a purposive sample of 18 Hispanic adolescents and young adults living with PHIV, who were recruited from two pediatric infectious disease centers in California, United States. Participants were eligible for the study if they were over the age of 15, self-identified as Hispanic, aware of their HIV status, and did not have cognitive impairments that would interfere with their ability to answer the interview questions. All participants were given the choice to have the interview conducted in Spanish or English. All chose to have the interview conducted in English. Participants and their parents or guardians (if the patient was under 18) gave written informed consent. The research was approved by the Institutional Review Board of Elon University (protocol code 12-009 and approved 16 June 2017).

### 2.2. Procedure

Interviews were held in person at each clinic in a private room and conducted by trained interviewers who were not part of the pediatric infectious disease teams. Interviews lasted approximately 45 min, were recorded, and subsequently transcribed. Transcripts were anonymized by removing any comments that may have compromised confidentiality. Upon completion, participants received a 20 USD gift card in appreciation of their participation.

### 2.3. Interview

The interview was based on questions developed by Fair and Albright (2012) and included open-ended questions about how HIV influences their relationships, sexuality, childbearing motivations, and career interests.

### 2.4. Analysis

After interviews were recorded and transcribed, they were entered into Dedoose, a qualitative data analysis software (Dedoose V6.1.18) [14]. The authors used a grounded theory approach to analyze the data, which employs an inductive strategy for examining the data for emergent themes [15,16]. Grounded theory is especially relevant for the current project since the participants assign meaning to their own experiences as opposed to the confirmation of existing hypotheses regarding their lives. Two authors independently read the interview transcripts in their entirety and following the traditions of the grounded theory method, analysis began with a process of open coding using Dedoose. The readers met frequently to discuss identified themes and to come to consensus on the coding. Once initial codes were identified, the authors searched for linkages across themes. Data saturation was achieved after the 15th interview. We followed the Consolidated Criteria for Reporting, a checklist for comprehensive reporting of qualitative studies [17].

## 3. Results

Through semi-structured interviews, this study explored the relationship experiences, childbearing intentions, and career intentions of Hispanic adolescents living with PHIV. Table 1 summarizes the demographic characteristics of participants (mean age = 20.8 years, range 15–29). Family country/territory of origin included Mexico (n = 6), several Central American countries (n = 9), Puerto Rico, (n = 1), and Cuba (n = 1). All but two of the participants were born in the United States and all received primary through secondary education in the United States. The majority (n = 11) were either in high school, had completed high school or had completed some college. Half were raised by a biological parent.

Relationships: “I would like us to be forever”.

Participants emphasized the importance of long-term intimate relationships. The majority were either in a committed relationship (n = 8) or married (n = 2) and 15 of the unmarried participants wished to be married in the future. Most indicated that living with HIV did not have a significant impact on the relationships they formed. For example, a 19-year-old male, responded, “It’s not hard, it’s normal”. Another noted, “I really don’t pay attention to it. … Yeah, it’s a second thought”. Nevertheless, almost every participant articulated concerns related to disclosure, fear of rejection, preventing transmission, and negotiating condom use. Five participants had never previously disclosed their HIV status to a partner.

Disclosure to partners was viewed as complex and adolescents articulated the challenges of knowing when it was the “right time” to tell a partner. A 17-year-old female explained, “You’re always scared that if you tell them at the wrong moment, then they’re just going to judge you. If you tell them on a right moment, then they’ll be like, ‘Okay, so why didn’t you tell me?’”.

Several participants shared painful stories of rejection, as indicated by a 20-year-old female who said, “People just assumed what they assumed. I’ve had people telling me I have it because I’m a whore or whatever. But they don’t really know me”. A 21-year-old explained, “I told one boyfriend. Then he didn’t want to be with me”. Many participants expressed that they did not feel in control of their disclosure and that their “secret” would spread without their consent.

“It’s because if you tell somebody straight out that did that, like you told them, then they might go and tell somebody else, and then that’s what you don’t want to happen. Then all this chaos happens”. (17-year-old female)

Participants also expressed fear that peers would not understand what it means to have acquired HIV perinatally, with one 19-year-old female stating, “I don’t know how they’ll react. Sometimes people don’t know what’s going on. Like people think it’s because you’re just having sex with other people …” One participant indicated that people believed he was infected due to a sexual relationship with his mother.

Several adolescents indicated that they remained with their current partner because they already knew and accepted their HIV status, even if their even if there were other issues within the relationship. A 19-year-old female said that even though her family believed she “could do better” than her current boyfriend, “we went through like, a lot”, and therefore, “forever he’ll be there for me”. A more extreme story was told by a 20-year-old who explained that the father of her 5-year-old child struggled with addiction to alcohol/drugs and had multiple incarcerations. She shared that he was angry when she disclosed her HIV status, introduced her to crystal methamphetamine, and did not take care of their child. She stated, “He always stay with me with bad or good things. I went with him because my mom threw all of my clothes out her house, and she didn’t want me there. I had to go to Leo [pseudonym]. He was there [for me]”. She says she has never considered dating someone else because she does not want to “end up by myself again”.

On the other hand, some participants also experienced acceptance upon disclosure. For example, one young woman received significant support from her partner:

“We were high school sweethearts. We’ve been together for three years. I told him when we were barely four months together. I called him and I was, ‘Okay, so yeah I have to tell you something’. He said, ‘What?’ I couldn’t tell him. I started crying before I even told him. He thought I was breaking up with him. Once I told him, I was, ‘I was born with HIV’. He’s, ‘Okay’. I’m, ‘Yeah’. He’s, ‘That’s it?’ I was, ‘Yeah, what do you mean that’s it?’ He’s, ‘Just take your medication and you’ll be fine’. I was super shocked. I was, ‘What?’ Why aren’t you disgusted or anything?’ He’s, ‘Why would I be disgusted? That’s fine. Just take your medication, keep your viral load down and you’ll be good’”.

An 18-year-old male recounted disclosing to his girlfriend, “She wasn’t mad at all. She was, well, I mean surprised maybe because she wasn’t expecting it. But I mean, she wasn’t mad at all, no negative comments or nothing, that she just supported me”. They are now married.

Seven participants were sexually active at the time of the interview, while ten reported being sexually active at some point in their lives; see Table 2 for relationship features. Eight participants reported feeling comfortable discussing sexual practices and issues related to their sexual relationships with their provider. Though providers proved to be a resource on sexual health, only seven participants recalled receiving sexual education as part of their middle or high school curriculum.

More than half the participants reported having used condoms at one point in their relationships. The majority were aware that condoms are the safest way to prevent HIV transmission, but condom use was inconsistent with five of the ten sexually active participants using condoms half the time or more. Partner preference was the most common reason for little or no condom use. A 22-year-old female expressed that sex and condoms were the biggest challenge in a relationship with HIV, stating, “There’s some guys who want to use it but some of them don’t. They just don’t know what the risk is and a lot of people don’t know that you infect each other back and forth”. Another female acknowledged the risk her partner was taking by not consistently using condoms, saying, “There were times that we didn’t use any because … I don’t know … he’s dumb. He’s so lucky. He’ll go every six months to get checked and then it’s negative. I was like, ‘Thank God, you’re not infected’”.

Childbearing: “As long as I can remember, I’ve wanted to be a mom, it’s just going to be difficult”.

Having a family of one’s own was noted as a meaningful part of life. Seven participants had children and the majority of participants desired to have children in the future. Motivations to bear children included partner preference, love of children, and a way to heal from past losses. Pressure from partners was the most common reason participants expressed a desire for children. A 22-year-old female explained, “the pressure is he wants to have a kid”. One third of participants cited a fondness and love for children, with a 19-year-old male stating, “kids are more cool, more adorable, cute and just beautiful”.

Finally, several participants asserted that raising children would allow them to make up for losses they experienced in their own childhoods. During the course of the interview, participants shared challenging childhood events that shaped their views on childbearing including the death of one or both parents, abuse, parental addiction, and/or incarceration, as well as instances of verbal abuse regarding their HIV status from other family members. Comments below underscore the strong desire to have children and be a good parent.

“I’ll give them what I didn’t have, probably not material but like the love and support”. (22-year-old female)

“I’d give them love, courage and good strength… something I never got”. (19-year-old male)

“I want to be there for [my daughter] so she can complete high school and then go to college—so she’s not like me”. (29-year-old female)

One participant explained that when she has children, she would have a new family and be able to leave her current one, “Right now, I have nothing … I want to be a parent. I want to have my own little family that I could tell my family bye-bye”.

Those who did not want more or future children expressed concerns about disclosing their HIV status to their child. Some participants without children expressed fear of mother-to-child transmission. One young woman stated, “I am afraid what the kid is going to have. My kid is going to have HIV”. Another participant’s girlfriend received a pregnancy test while he was being interviewed. He said they would be excited if she were pregnant, but stated, “…there is a high risk that I could pass it to her and then maybe pass it to the baby. We’re totally afraid, I mean, I’m freaking about that all HIV thing”.

One young woman found maintaining a lasting supportive relationship difficult, which in turn affected her daughter and reduced her interest in having additional children:

She [the child] has issues and I think it’s because her dad’s incarcerated and then my ex-boyfriend that was supposed to be a father figure, like he would stay there, take her to the park, babysitting while I was working, everything, and then just for him to walk out on her life, it was devastating for her.

Careers: “I want to make a difference. I want to be that person for a child, cause it’s really hard”.

When asked about future intentions regarding school and careers, the majority of AYA described a plan for future education and/or a specific job or career path. Participants expressed interest in medical fields, teaching, and research as well as the entertainment and culinary industries. Others were uncertain about their future education or employment plans.

Most participants said their HIV status did not affect their plans for education, career aspirations, or other aspects of their future. One attributed this to the fact that he has lived with HIV since birth, explaining, “They don’t really affect me, because I’ve been living with it my whole life. If it was something new, that would be something different. I’ve been living with it now since … I was born with it”. However, two participants responded that their health has, in the past, negatively interfered with jobs. One participant elaborated, “besides the fact that I get sick a lot. That kind of holds me back, but I don’t try to let it get to me”. Another added that in addition to his own health, he worried about the possibility of transmitting HIV while working in a kitchen:

“Because I get sick, yes, that’s one of the issues… because I’m not stable right now. I’m be getting sick a lot … and calling off, that I’m get fired. I don’t want to work around food because I’m afraid I might cut myself … and because for some reason I cough a lot. I know I cannot work around food…because just because I have to grab a knife, I might poke myself probably the blood ends in the food and I don’t know [if] that’s going to affect somebody else”. (19-year-old male)

Similarly, another participant expressed concern that her HIV status would be a problem in her desired field of nursing. “[My HIV status] has affected me negatively … like if I would be accepted type of thing. When I think about it, like maybe if I’m a nurse, but maybe they don’t want a nurse like me working there”.

More than HIV, participant’s pregnancy and parenthood affected future school and career intentions. Four participants stated that having children made it more difficult or impossible to complete high school and find a job, which changed their thoughts about future careers after having children. One young mother stated, “Well, my dream was to go to the army, but now it’s like kind of don’t want to go no more. Because …I can’t leave my baby like that. So, it’s for me, just like, I’m going to put that aside”.

Participants with a clear idea of their future aspirations most often had a source of support that included health care providers, close family, or former mentors/teachers. One participant noted that her choice to become a medical assistant was heavily influenced by living with PHIV:

“Just growing up with it and having a great support system, whether it was camp or social workers, just knowing that I could go and talk to somebody when I needed it, I want to be that person for a child, cause it’s really hard”. (29-year-old female)

Another 20-year-old male participant indicated his choice of becoming a teaching assistant was influenced by his high school principal, “I like the way he taught us and how he was straight up with the students”.

Many adolescents indicated they had discussed future education and career intentions with providers in the clinic and received encouragement and support. “They tell me to keep it up and to follow my dream”, said a young woman. Another teen who wants to go to nursing school said, “I do tell them about my future plans … They’ve asked me and they think that’s a good idea too, for me to go back to school and finish what I didn’t finish …. I mostly sometimes ask the nurses what schools they went to so I could be more aware and have some knowledge of what steps I have to take, so just to ask”.

However, some participants desired even more specific advice, as well as information on financial resources. A participant who hoped to be a researcher indicated she wanted to know “The steps you have to take or what classes”. Another added, “My social worker has told me that that they are here for me. That they’ll help in what they can, but I get embarrassed of asking them like I don’t have money for this, I don’t have that”.

## 4. Discussion

This study sought to understand the lived experiences of Hispanic AYA living with PHIV specifically focused on their personal and sexual relationships, childbearing decisions, and career aspirations. To our knowledge, this is one of the first studies to focus solely on this unique population. Hispanic AYA living with PHIV reported similar experiences as other youth with PHIV, such as challenges and fears related to disclosure, rejection, and risk of transmission regarding romantic and sexual relationships [4,5]. Although less than half of participants recalled receiving a sexual education course in school, quite a few indicated they felt comfortable asking sexuality-related questions of their health care providers. Previous research suggests that providers who care for AYA with PHIV spend a significant amount of time on HIV transmission prevention [18]. Participants expressed concerns that disclosure would lead to assumptions of risky sexual behavior, something that goes against Hispanic cultural values [11]. Given that disclosure was noted as a primary relationship stressor, it may be helpful for providers to process their concerns, discuss culturally appropriate strategies to support healthy disclosure.

Childbearing desires and intentions were high within the study population, confirming previous research on adolescents living with PHIV [19]. In a qualitative study of twenty AYA living with PHIV in Puerto Rico, Silva-Suárez et al. also found that most wished to become parents [20]. Fair and Albright noted the desire to seek healing from early childhood losses and improve the world for the next generation were common childbearing motivations among their study population of youth with PHIV [21].

Similar to findings by Silva-Suárez et al. [19] and Fair et al. [22], health care providers were seen as a significant source of support. Support ranged from simply being a positive role model to providing specific advice for career intentions. Complications due to HIV can adversely influence education and employment opportunities [23]. However, most participants did not indicate that HIV had interrupted their intended plans for pursuing further education or employment. Indeed, study participants were aware that their health care providers were a source of information and support not otherwise available to their same-age peers. However, several participants held misconceptions regarding the risk of HIV transmission in the workplace and may benefit from continued education surrounding the nature of HIV transmission. Indeed, infectious disease health care providers are in a unique position to offer ongoing education related to transmission risk and prevention of unintended pregnancy [18]. Culturally sensitive health promotion education tailored specifically for youth with PHIV can help offset the varying quality of sex education offered in most public high schools [24].

## 5. Limitations

The relatively small sample size and geographic location limits generalizability. Furthermore, all participants were connected to care and findings may not be applicable to adolescents not receiving medical care. Future research should further explore the more nuanced cultural experience of Hispanic adolescents with PHIV. Due to limited sample size, we could not report results by country/territory of origin, though we know previous research has noted the limitations of combining all Hispanic people into a large group [25]. However, our sample was too small to distinguish any differences in cultural norms or values. Furthermore, we did not collect information on the race/ethnicity of sexual partners, which could also impact sexual practices and childbearing plans. Finally, all data are qualitative and based on self-report and researchers did not have access to medical charts.

## 6. Conclusions

Despite limitations, this exploratory study contributes to the relatively new body of literature on Hispanic AYA with PHIV. Hispanic AYA are engaging in romantic and sexual relationships at similar rates as the general population of AYA with PHIV and typically developing adolescents. Comprehensive sexual health programs and open discussion of sexual practices would prove useful for this population especially as they seek accepting long-term partners. Economic empowerment, culturally competent care, and connection to emotional support resources could facilitate the well-being of Hispanic AYA living with PHIV. Future research should include longitudinal studies that track the well-being of this vulnerable population over time in order to identify interventions that will support successful adaption to adulthood.

## Figures and Tables

**Table 1 ijerph-20-02996-t001:** Demographics.

N = 18	Descriptor	No.	%
Sex	Male	6	33.3
	Female	12	66.7
Family country of origin			
	Mexico	6	33.3
	Honduras	4	22.2
	Guatemala	3	16.7
	Puerto Rico	2	11.1
	El Salvador	1	5.6
	Columbia	1	5.6
	Cuba	1	5.6
Have child		7	38.9
Current Living Situation	Parent	7	38.9
	Family (not parents)	4	22.2
	Homeless	1	5.6
	Foster Care	1	5.6
	Partner	4	22.2
	Independent	1	5.6
Education	Some College	1	5.6
	Completed High School	4	22.2
	Currently in High School	6	33.3
	Some High School	7	38.9
Employment	Unemployed	8	44.4
	Current Student	6	33.3
	Employed	4	22.2
Biological Mother Alive	Yes	10	55.6
	No	8	44.4
Raised By	Biological Parent	9	50.0
	Other Family	7	38.9
	Adoptive Parents	2	11.1

**Table 2 ijerph-20-02996-t002:** Relationships.

	Descriptor	No.	%
Relationship Status	Single	5	27.8
	Casual relationship	2	11.1
	Committed relationship	8	44.4
	Married	2	11.1
Previously Disclosed HIV status to a partner	Yes	12	66.7
Sexually Active with Current Partner	Yes	7	38.9
Desire Marriage	Yes	17	94.4
Sexually Active past 6 mos.	Yes	10	55.6
Contraception use past 6 mos.	Yes	8	44.4
	No	3	16.7
	N/A	7	38.9
Frequency of Condom Use (among sexually active)	Always	5	50
	Inconsistent	2	20
	Never	3	30
Received Sexual Education	School	7	38.9
	Health care provider	8	44.4

## Data Availability

Available upon request.

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
