# Peer review of "“We Are Not Different than Others”: A Qualitative Study of the Lived Experience of Hispanic Adolescents and Young Adults Living with Perinatally Acquired HIV"

_ijerph, 2023, doi:10.3390/ijerph20042996_

Round 1
Reviewer 1 Report
This is an interesting and well written paper. However, the following issues needs to be addressed.
1. In the method section kindly specify whether saturation was achieved or not.
2. Did the authors follow the COREQ guidelines for reporting qualitative studies? If yes, kindly report it and include the report. If not kindly state it in the limitations.
3. The discussion can be further improved. In addition, also include references of quantitative studies to buttress the qualitative findings.
Author Response
Thank you for reviewing our work. We have responded to comments below:
Reviewer 1
This is an interesting and well written paper. However, the following issues needs to be addressed.
- In the method section kindly specify whether saturation was achieved or not.
- Saturation was achieved after the 15th interview and that is now included in the methods section.
- Did the authors follow the COREQ guidelines for reporting qualitative studies? If yes, kindly report it and include the report. If not kindly state it in the limitations.
- Thank you for this suggestion. We did follow the COREQ guidelines and have integrated that into the methods section.
- The discussion can be further improved. In addition, also include references of quantitative studies to buttress the qualitative findings.
- We have made better linkages between the discussion and the introduction. To our knowledge, there are no quantitative studies that focus exclusively on this unique population.
Reviewer 2 Report
The manuscript is well written, but I suggest the following to improve the quality of the final article
i) The authors have made a good attempt at situating the paper in past research. However, I feel a more detailed discussion of what has been done by previous researchers could be beneficial in terms of providing more conceptual background.
ii) I wish there was a theory/theoretical framework to underpin this article.
iii) A section where the findings are discussed could also be helpful in telling the reader how these findings are linked with the literature reviewed and presented in the article's introduction section.
Author Response
Thank you for reviewing our work. We have responded to comments below:
- i) The authors have made a good attempt at situating the paper in past research. However, I feel a more detailed discussion of what has been done by previous researchers could be beneficial in terms of providing more conceptual background.
- Thank you for this suggestion. To our knowledge, we have cited all the literature available on this population. Further we included information about AYA with PHIV in general as well as Hispanic youth.
- ii) I wish there was a theory/theoretical framework to underpin this article.
- This study was based upon grounded theory where participants assign meaning to their own experiences rather than confirming or refuting a prior hypothesis.
iii) A section where the findings are discussed could also be helpful in telling the reader how these findings are linked with the literature reviewed and presented in the article's introduction section.
- We appreciate this suggestion and have made more direct connections between the findings and the introduction in the discussion section.